# Risk Factors for Surgical Treatment of Endometrial Cancer Using Traditional and Laparoscopic Methods

**DOI:** 10.3390/jcm10030429

**Published:** 2021-01-22

**Authors:** Sławomir M. Januszek, Barbara Wita-Popow, Marta Kluz, Magdalena Janowska, Rafał Januszek, Andrzej Wróbel, Artur Rogowski, Krzysztof P. Malinowski, Tomasz Zuzak, Tomasz Kluz

**Affiliations:** 1Department of Gynecology and Obstetrics, Fryderyk Chopin University Hospital No. 1, 35-055 Rzeszów, Poland; wita.barbara@gmail.com (B.W.-P.); m.janowska@wp.pl (M.J.); tomasz.zuzak@gmail.com (T.Z.); jtkluz@interia.pl (T.K.); 2Department of Pathology, Fryderyk Chopin University Hospital No. 1, 35-055 Rzeszów, Poland; marta.kluz@interia.pl; 3Department of Clinical Rehabilitation, University of Physical Education, 31-571 Kraków, Poland; jaanraf@interia.pl; 4Second Department of Gynecology, Medical University of Lublin, Jaczewskiego 8, 20-954 Lublin, Poland; wrobelandrzej@yahoo.com; 5Faculty of Medicine, Collegium Medicum, Cardinal Stefan Wyszyński University in Warsaw, 01-938 Warsaw, Poland; arogowski@op.pl; 6Department of Obstetrics and Gynecology, Mother and Child Institute, 01-211 Warsaw, Poland; 7Faculty of Health Sciences, A Institute of Public Health, Jagiellonian University Medical College, 31126 Kraków, Poland; krzysztof.piotr.malinowski@gmail.com; 8Department of Gynecology and Obstetrics, Institute of Medical Sciences, Medical College of Rzeszow University, 35-310 Rzeszów, Poland

**Keywords:** endometrial cancer, total laparoscopic hysterectomy, surgical treatment, sentinel node procedure lymphadenectomy, abdominal obesity, perioperative outcomes

## Abstract

Surgical treatment is the most important part of therapy for endometrial cancer. The aim of the study was to define factors having the most significant impact on surgical treatment of endometrial cancer when using traditional and laparoscopic methods. In the study, we evaluated 75 females who were treated for endometrial cancer via laparoscopic surgery in 2019 and used a historical control of 70 patients treated by laparotomy in 2011. The evaluated risk factors included the method of surgery, type of lymphadenectomy, patient’s age, various obesity parameters, histological grading, cancer clinical staging, pelvic dimensions, previous abdominal surgeries, comorbidities, and number of deliveries. The duration of hospitalization, operation time, loss of hemoglobin, and procedure-related complications were used as parameters of perioperative outcomes. Multivariable linear regression analysis confirmed the following factors as being predictors of worse perioperative outcomes: laparotomy, abdominal obesity (waist circumstance and waist-to-hip ratio), range of lymphadenectomy, prior abdominal surgeries, and larger pelvic dimensions. Abdominal obesity is a significant risk factor in the treatment of endometrial cancer. Laparotomy continues to be utilized frequently in the management of endometrial cancer in Poland as well as elsewhere, and adopting a minimally invasive approach is likely to be beneficial for patient outcome.

## 1. Introduction

Endometrial cancer is the most common gynecological cancer in developed countries, and the fourth most common cancer in women [1]. During the last three decades, the rates of uterine cancer have increased by over 50%. This increase in incidence has been largely attributable to elevated rates of obesity [2]. The most significant treatment for endometrial cancer is surgery. The factors modifying the course of surgical treatment of endometrial cancer include method of surgery, various obesity parameters, histological type of the cancer, cancer clinical stage, type of lymphadenectomy, patient’ age, comorbidities, pelvic anatomical parameters, and the number of previous abdominal surgeries.

In randomized controlled trials, it has been shown that laparoscopy tends to be the preferred treatment for early stage endometrial cancer in obese patients, as it is associated with less postoperative pain, shorter hospitalization, and a faster return to daily activities compared with open surgery [3,4]. In this review, it was also confirmed that mini-invasive surgery seems to be safe in the treatment of patients with high-risk endometrial cancer, showing better perioperative and postoperative results, and maintaining comparable oncological outcomes to open surgery [5]. About twenty years ago, women with endometrial cancer qualifying for laparoscopic management had a substantially lower body mass index (BMI) than patients operated on using traditional procedures because obesity was a relative contraindication to laparoscopy [6]. In further studies, it was concluded that laparoscopic procedures in obese patients with endometrial cancer are a safe and feasible alternative treatment to laparotomy [7]. Nevertheless, it has been noted that obesity is associated with higher mortality from causes other than endometrial cancer or disease recurrence [3]. The confirmed risk factor is the clinical staging of cancer due to various reasons, such us higher risk of bleeding, damage to adjacent organs, and infection. It has also been shown that a greater extent of lymphadenectomy is a risk factor for perioperative complications [8]. Patient’s age and comorbidities are recognized risk factors in practically all areas of surgical treatment. Previous abdominal surgeries are often associated with intra-abdominal adhesions, which contribute to longer surgery time, greater bleeding, infections, and other perioperative complications. Bone structure may limit the surgical field in a narrow pelvis, and reduce the precision of the surgeon’s movements in a pelvis that is extremely deep. The influence of pelvic dimensions on the course of surgical treatment of prostate cancer was investigated [9]. It was also found that pelvic dimensions were predictors for anastomotic leak in rectal cancer patients who underwent anterior resection [10]. Observations were carried out on the group of patients treated with laparotomy in 2011 and on the group of patients treated via laparoscopy in 2019. Preoperative clinical differences between the groups of patients treated with the classical method and those treated with laparoscopy may also result from the difference in the time between conducted observations. The aim of the study is to describe the risk factors that have a significant impact on the course of surgical treatment of endometrial cancer, and to report methods improving surgical management.

## 2. Materials and Methods

The study included 145 patients with endometrioid cancer who qualified for surgery based on histopathological results, physical examination, transvaginal ultrasound, laboratory tests, and computed tomography of abdominal and pelvic cavity, in some cases. Preoperative histopathological diagnosis was made on the basis of tissues obtained mainly by targeted biopsy during hysteroscopy, and in some cases, by curettage (diagnosed most often at regional hospitals). The research was an observational prospective study of surgical treatment for endometrioid cancer performed at an oncological gynecology center. The operations were performed under the supervision of three specialists in oncological gynecology, all with 20–30 years of experience in surgical gynecology, including approximately 20 years in laparoscopic surgery. This was a non-randomized, prospective cohort study followed by observation limited to hospitalization duration and 30 days post-discharge. During hospitalization, the loss of hemoglobin related to the procedure and the duration of the procedure and hospitalization were assessed, while in the periprocedural period and for up to 30 days following discharge, the presence of postoperative complications was assessed, such as gastrointestinal obstruction or infection of the postoperative wound with impaired healing. The study was conducted in two stages. During the first stage in 2011, approximately 90% of endometrial cancer cases were treated with open surgery in our hospital. The course of treatment for 70 patients with endometrial cancer operated on using the traditional method was analyzed. In less than a decade, this proportion has practically reversed, and now, the vast majority of endometrial cancer cases are qualified for laparoscopic treatment. The second stage of research was carried out in 2019, when 75 patients approved for laparoscopic treatment of endometrial cancer were included in the study. Before surgery, thorough clinical assessment and physical examination were performed. The clinical interview included age, number of deliveries, education in years, previous abdominal surgeries, and comorbidities. During the clinical examination, height and body mass, waist and hip circumference in centimeters using a tape measure, and pelvic bone dimensions using pelvic meter were measured. Body mass index (BMI) and waist-to-hip ratio (WHR) were also calculated during preoperative examination. The protocol is compliant with the Declaration of Helsinki and has been approved by the local ethics committee at the University of Rzeszów (No. 4/12/2011).

Patients were qualified for the treatment according to applicable Polish guidelines [11]. Open surgical treatment included hysterectomy with bilateral salpingo-oophorectomy, systemic pelvic and para-aortic lymphadenectomy (Histological grading -G1/G2 and myometrial infiltration >50%, G3). There were no cases of serous carcinoma or carcinosarcoma included in the study, in which omentectomy should be performed. Intraoperatively, myometrial infiltration was assessed macroscopically after hysterectomy. The average number of lymph nodes removed during pelvic lymphadenectomy in patient with lymphadenopathy during open surgery was 19. Laparoscopic treatment included sentinel lymph node procedure, total hysterectomy with bilateral salpingo-oophorectomy, and systemic pelvic and para-aortic lymphadenectomy. The laparoscopic method is suitable for sentinel detection in surgical treatment of endometrial cancer, due to the increased magnification and illumination of the operating field [12]. The sentinel node procedure was performed in 45.3% of patients treated with laparoscopy. In patients undergoing sentinel node procedure, 2 mL (1 mg/mL) of dye was injected into the cervical stroma, divided between superficial 1–3 mm injection and deep 10–20 mm injection at 3 and 9 o’clock before placing the manipulator into the uterus. Retroperitoneal spaces were explored prior to hysterectomy, and fluorescence imaging was used to detect the sentinel node. The identified sentinel nodes were removed and subjected to intraoperative histopathological examination, followed by routine hematoxylin and eosin staining. The patients then underwent hysterectomy and bilateral salpingo-oophorectomy. According to the sentinel lymph node procedure, all suspect lymph nodes were removed during surgery, regardless of mapping. If no lymph node was stained (no mapping) on one side of the pelvis, a homologous unilateral pelvic lymphadenectomy was performed. Para-aortic lymphadenectomy was carried out at the surgeon’s discretion. In the absence of the sentinel node procedure, the patient underwent hysterectomy and bilateral salpingo-oophorectomy. Intraoperatively, myometrial infiltration was assessed macroscopically after hysterectomy in all cases, regardless of the sentinel lymph node procedure. Subsequently, the patients underwent systematic pelvic and para-aortic lymphadenectomy in some cases of moderate, and in all cases of high risk. In cases concerning intermediate risk of G1/G2 endometrioid carcinoma and myometrial infiltration (MI) > 50% or G3 endometrioid carcinoma and MI < 50%, lymphadenectomy was considered. In high-risk cases of G3 endometrioid carcinoma and MI > 50%, and nonendometrioid carcinoma and in all cases of clinical stage II, IIIA, and IIIB, the lymphadenectomy was performed [11]. The mean number of lymph nodes removed by laparoscopy in patients with lymphadenopathy was 16. Complete staging according to the FIGO (International Federation of Gynecology and Obstetrics) classification was established on the basis of postoperative histopathological results of all tissues removed during surgery.

The range of lymphadenectomy has been divided into three classes for statistical purposes: class 1—selective lymphadenectomy in laparotomy or sentinel node procedure in laparoscopy, class 2—pelvic lymphadenectomy, and class 3—pelvic and para-aortic lymphadenectomy. Similarly, for statistical purposes, numbers were assigned to stages of endometrial cancer according to FIGO (International Federation of Gynecology and Obstetrics): 1-Ia, 2-Ib, 3-II, 4-IIIa, 5-IIIb, 6-IIIc1, 7-IIIc2, 8-IVa, and 9-IVb.

### 2.1. Study Endpoints

In the perioperative period, the parameters considered as determinants regarding hospital outcomes of surgery were monitored: duration of the procedure (in minutes), the loss of hemoglobin—the difference in the serum concentration before surgery and on the second day after surgery (gram/decyliter, occurrence of complications, and hospitalization duration (days).

### 2.2. Statistical Analysis

Continuous variables are expressed as mean ± standard deviation. Categorical variables are introduced as numbers and percentages. Normality was assessed with the Shapiro–Wilk test. The Mann–Whitney continuous variables in selected groups of patients were compared using the Welch test, Kruskal–Wallis test, and the Student’s *t*-test, where applicable. Categorical variables were compared with the use of the chi-square test. Univariate and multivariate linear regression analyses were used to find significant predictors of the selected study endpoints. All potential predictors with clinical values were included in multiple regression modelling. The best models for prediction of hospitalization and procedure duration, as well as hemoglobin loss were obtained using backward elimination with Akaike Information Criterion as a target. If two variables were highly correlated (correlation coefficient >0.7), the strongest predictor was selected. The final results were presented as point estimates with 95% confidence intervals and *p*-values. R2 coefficients were calculated. Bootstrap model validation was performed with 1000 iterations. Model assessment was carried out by the examination of residuals. Statistical analysis was executed in R version 3.6.2 (R Foundation for Statistical Computing, Vienna, Austria, 2019). The *p*-value of <0.05 was considered statistically significant.

## 3. Results

### 3.1. General Characteristics

There was no significant difference in the mean age of study participants between the laparoscopic and traditional groups (62.2 ± 8.7 years vs. 61.3 ± 10.4 years, *p* = 0.55) (Table 1).

In the group of patients treated with laparoscopy and laparotomy, there was also no significant difference in the mean number of deliveries (2.6 ± 1.2 vs. 2.2 ± 1.2, *p* = 0.13), number of prior abdominal surgeries (0.65 ± 0.8 vs. 0.57 ± 0.6), or concomitant diseases (Table 2).

### 3.2. Selected Anthropometric and Anatomical Measurements

There was no significant difference in the mean body mass index value between both assessed groups of patients (32.2 ± 8.0 kg/m^2^ vs. 32.1 ± 7.3 kg/m^2^, *p* = 0.73). Although there was a significant difference in WHR, which was greater among patients treated with laparoscopy compared with the group undergoing traditional surgery (0.94 ± 0.1 vs. 0.92 ± 0.1, *p* = 0.04), this difference did not appear to be clinically relevant. There were also no significant differences in external conjugate (*p* = 0.21), interspinous (*p* = 0.15), intercristal (*p* = 0.56), and intertrochanteric diameters (*p* = 0.15) between the groups of patients treated by laparoscopy and open surgery. These and other anthropometric measurements are presented in Table 1.

### 3.3. Tumor Staging and Grading

The extent of lymphadenectomy was significantly greater in the laparoscopy group when compared to the laparotomy group (1.6 ± 0.7 vs. 1.1 ± 0.9, *p* = 0.007). This was mainly attributed to the higher percentage of patients from the first and third classes in the laparoscopy group when compared to the traditional procedure group (*p* < 0.001). However, the more frequent lymphadenectomy did not contribute to the increased detection of lymph node metastases (no statistically significant differences in stage IIIC). When considering staging according to the FIGO classification, it was also greater in the laparoscopic group when compared to the group subjected to the traditional method (2.3 ± 1.3 vs. 2.0 ± 1.5, *p* = 0.01). In the postoperative histopathological results, patients treated with laparoscopy were statistically more often diagnosed as IB stage, and less often as stage IA, according to FIGO (Table 3). There were no significant differences between the analyzed groups in histological grading, as presented in Table 3.

### 3.4. Procedure-Related Complications

There were ten perioperative complications. In the group of patients treated with laparoscopy, five complications occurred: infection of the postoperative wound with impaired healing in one patient, infection of vaginal wound requiring prolonged systemic antimicrobial therapy in two patients, conversion to open surgery due to abdominal obesity and preperitoneal entry and insufflation in one patient, and omental damage and bleeding in one patient. All of these complications occurred in obese females. There were also five perioperative complications in the group of patients treated with laparotomy: infection of the postoperative wound with impaired healing in two patients, intraoperative bladder injury in one patient, gastrointestinal obstruction in one case, and intraoperative bleeding with the need of internal iliac artery ligation in one patient. There was no significant difference in the procedure-related complication frequency between both groups of patients (7.1% vs. 6.7%, *p* = 0.91).

### 3.5. Duration of Hospitalization

The duration of hospitalization in patients treated with laparotomy was significantly longer compared with laparoscopy (11.3 ± 3.1 days vs. 6.5 ± 1.7 days, *p* < 0.001). These durations appear longer than in some other centers [13,14]. The longer hospitalization time in this study can be explained by local specificity of outpatient care, which is ineffective both in the period of preoperative diagnostics and in postoperative supervision. Considering the overall group of patients, duration of hospitalization was significantly positively correlated with body mass, BMI, waist and hip circumference, and external conjugate diameter (Table 4).

Multivariable linear regression analysis allowed the following to be confirmed as the most significant predictors of longer hospitalization duration: open surgery, waist circumference, greater number of prior abdominal surgeries, smaller number of deliveries, and duration of operation (Figure 1).

### 3.6. Duration of Surgery

The mean duration of surgery was longer in patients treated with laparoscopy. However, this was without statistical significance (91.4 ± 18.2 min vs. 86.1 ± 20.4 min, *p* = 0.09). In the overall group of patients, there was significant positive correlation between the duration of operation and the following: body mass, BMI, waist and hip circumference, WHR, intertrochanteric diameter, and interspinous, intercristal, and external conjugate diameters. Multivariable linear regression analysis confirmed the following to be significant predictors of longer surgery duration: laparoscopy, waist circumference and intertrochanteric diameter, higher staging according to FIGO, and greater number of deliveries (Figure 2). It is worth noting that the operation time also depends on other factors not taken into account in the study, including experience of the operator and available equipment.

### 3.7. Procedure-Related Hemoglobin Loss

The mean hemoglobin loss during the periprocedural period was significantly greater in the group of patients treated with laparotomy compared with those subjected to laparoscopic treatment (2.25 ± 0.9 g/dL vs. 1.0 ± 0.2 g/dL, *p* < 0.001). In the overall group of patients, procedure-related hemoglobin was significantly and positively correlated with body mass, BMI, waist and hip circumference, as well as external conjugate diameter. Multivariable linear regression analysis confirmed the following significant predictors of greater procedure-related hemoglobin loss: traditional mode of hysterectomy, WHR, greater range of lymphadenectomy, and external conjugate diameter (Figure 3).

## 4. Discussion

### 4.1. Risk Factors for Surgical Treatment

Perioperative outcomes affect both the mental and physical condition of patients treated for endometrial cancer. Longer operation and hospitalization duration, as well as greater loss of hemoglobin prolong convalescence, which significantly affects the quality of life of patients, and may also delay complementary treatment. Worse periprocedural outcomes also raise the overall cost of treatment. When estimating the perioperative risk, one should be guided by the most adequate indicators. In our study, it was indicated that the most adequate predictors of perioperative outcomes among obesity parameters are parameters of visceral obesity: waist circumstance and waist-to-hip ratio. This study shows that in practice, these parameters should be used more often than BMI or body mass when estimating perioperative risk. In assessing perioperative risk, some pelvic dimensions and those from previous abdominal operations could also be taken into account. Despite the limitations of the control group, the authors of the study suggest that patients with certain risk factors, such as abdominal obesity, should be considered for laparoscopy more frequently. The relationship between the extent of lymphadenectomy and periprocedural outcomes also suggests more frequent implementation of the lymph node procedure among patients with higher perioperative risk.

### 4.2. Clinical Staging

In the last decade, due to improvement of the laparoscopic method of treatment for endometrial cancer and the development of surgeons’ skills, both obese patients and those at a higher clinical stage of cancer were more frequently qualified for laparoscopic treatment. As a result of improvement in the laparoscopic treatment method, in 2019, more than 80% of patients with endometrial cancer were treated by laparoscopy at our clinic. As a consequence of these changes, in the observational study of patients treated with the traditional and laparoscopic methods, an analogous group with similar clinical characteristics was obtained. However, patients treated with laparoscopy were statistically more often diagnosed as IB stage, and less often as stage IA, according to FIGO (Table 2). These differences may be caused by the fact that, over time, patients at higher stages, according to FIGO, reached our clinic, where they were qualified for laparoscopic treatment. Lymphadenectomy was performed more frequently in patients treated by laparoscopy, which was a consequence of more frequent infiltration of the myometrium deeper than 50% of its thickness predicted preoperatively (in ultrasound examination or computed tomography), and assessed macroscopically after hysterectomy during operation.

### 4.3. Obesity

In the presented study, multivariate regression indicated that the parameters of visceral obesity determine hospitalization time (waist circumference), operation duration (waist-to-hip ratio), and loss of hemoglobin (waist circumference). Chronic inflammation and metabolic disorders, more common in abdominal obesity, may also contribute to infections or longer healing of wounds, as well as circulatory and respiratory disorders [15]. Ucella et al. [16] also showed that obese patients have a higher frequency of complications compared with those non-obese, regardless of the treatment method. We recorded five perioperative complications during laparoscopy and five during open surgery, which occurred only in obese patients. These authors observed a declined percentage of patients undergoing lymphadenectomy with a BMI >40 [16], while in our study, we noted a greater mean extent of lymphadenectomy in patients treated with laparoscopy. In a retrospective study conducted in 2005–2009 among patients treated for endometrial cancer via laparoscopy and open surgery, a higher number of postoperative complications were found in patients with morbid obesity [17]. In the study, it was also shown that in patients with a BMI < 40, the number of complications does not differ depending on the degree of obesity. The authors concluded that obesity is not an independent risk factor for postoperative complications, which significantly depended on comorbidities of obesity [17]. However, in that study, the only obesity parameter was BMI, while in our study, we showed that abdominal obesity may be a more significant predictor of perioperative outcomes. Among patients treated with the traditional method, the wound surface is naturally larger in obese patients, especially among those with abdominal obesity. Minimal abdominal invasive surgery can be also a challenging and hazardous procedure in overweight or obese patients, especially in endometrial cancer patients with abdominal obesity [18]. Abdominal obesity also increases the risk of preperitoneal entry and conversion to laparotomy, because in patients with abdominal obesity, the distance from the skin to peritoneum is larger, which may complicate initial trocar introduction [19]. In various studies, including the research by Walker et al., it has been shown that there is a significant correlation between increasing BMI values in patients and increasing conversion rate to laparotomy, reaching over 50% in patients with a BMI > 35 kg/m^2^ [20].

### 4.4. Pelvic Dimensions

There is an interesting relationship between the dimensions of the pelvic bones and the duration of surgery, as well as hemoglobin loss. This dependence may be related to the fact that larger distances between the pelvic bones reduce the precision of movements, which is especially visible in laparoscopy when it is necessary to operate with longer instruments. Moreover, larger pelvic dimensions are more common in obese women.

### 4.5. Previous Abdominal Surgeries and Deliveries

Multiple linear regression also indicates that a greater number of previous abdominal surgeries and longer surgery time are predictors of extended hospitalization. The greater number of deliveries was associated with a shorter hospitalization, which may be due to a lower number of comorbidities in patients with such a history. On the other hand, a greater number of deliveries correlated with longer operation time, which could be associated with a different structure of the pelvis, or more adhesions after caesarean sections.

### 4.6. Method of Surgery

It is worth noting that the choice of the traditional surgical method is a predictor of longer hospitalization duration and greater loss of hemoglobin. In the presented study, there was no significant difference in the incidence of postoperative complications between the group of patients undergoing traditional surgery and those undergoing laparoscopy. Ucella et al. [16], in a multicenter, retrospective study, examined 1266 patients, including 746 treated with laparoscopy and 502 who underwent open surgery. In this trial, longer postoperative hospitalization, higher frequency of blood transfusions, and a higher incidence of postoperative complications were shown in patients treated with open surgery compared with the group of patients treated via laparoscopy. In our study, we also found similar correlations between the method of surgery and the time of hospitalization, as well as perioperative blood loss [14]. However, in the present study, we considered total hospitalization time and the difference in hemoglobin concentration one day before the surgery and on the second day following the surgery as parameters of in-hospital outcomes. We did not confirm a statistically significant difference in the frequency of complications for both groups, probably due to smaller number of patients included in our study. Mahdi et al. [17] also showed that the incidence of perioperative complications is higher in patients treated with open surgery. Moreover, complications occurred more frequently in patients with a higher degree of obesity, while in patients treated with laparoscopy, the incidence of complications did not change significantly with increasing obesity. The authors suggest that perioperative risk may be attenuated by the use of a minimally invasive approach [17]. In another retrospective study among 627 patients surgically managed for endometrial cancer between 2006 and 2015, it was confirmed that morbid obesity patients treated with open surgery were at the highest risk of postoperative complications [21]. The authors of this article also concluded that laparoscopic surgery may prevent the majority of postoperative complications in morbidly obese patients [21]. It was also confirmed in further studies that laparoscopy is safe in those morbidly obese [22], and appears to be safe even for the treatment of high-risk endometrial cancer patients, showing better perioperative and postoperative outcomes as well as comparable oncological end results with open surgery [5]. In large randomized controlled trials, laparoscopic hysterectomy and bilateral salpingo-oophorectomy have been shown to be a preferable technique in women with endometrial cancer due to lower estimated blood loss, shorter hospital stay, and lower incidence of perioperative complications [23]. Additionally, in a systematic review of randomized trials, in which laparoscopy and laparotomy were compared for early stage endometrial cancer, the authors found no significant differences in the risk of death among women who underwent laparoscopy and those who were subjected to laparotomy [24,25]. The results of one meta-analysis (*n* = 313) also confirmed that women in the laparoscopy group lost significantly less blood than women in the laparotomy group [26]. Ruan et al., described a significantly higher percentage of local complications in postoperative wounds in the case of laparotomy, as opposed to the laparoscopic method. In this study, following laparoscopy, statistically significant lower values were obtained for scales of postoperative pain assessment [27]. The subject literature provides data on the results of traditional surgery compared to the laparoscopic method in patients above the age of 60. There is a statistically significant increase in the number of surgical and internal complications, for example, the need for general antibiotic therapy, intestinal obstruction, pneumonia, and venous thrombosis, in the group operated on via the traditional method. In this study, the overall greater disadvantage of the traditional approach compared with laparoscopy was demonstrated in the older group of patients [28].

### 4.7. Lymphadenectomy

It was also confirmed by multivariate linear regression analysis that the extent of lymphadenectomy is a significant factor increasing hemoglobin loss. The lowest hemoglobin loss was found during sentinel node surgery using laparoscopy, and the highest during pelvic and para-aortic lymphadenectomy in laparotomy. Laparoscopic lymphadenectomy does not show less oncological efficacy than open surgery, also in the treatment of patients with early [29], moderate-, and high-risk endometrial cancer [28]. Pineda et al. [1], through a retrospective study, showed that use of the sentinel node procedure improves staging in endometrial cancer by low-volume metastasis detection, and it should be implemented in treatment of endometrial cancer. In our study, it was not confirmed that the sentinel node procedure contributes to increased detection of metastases to the lymph nodes (small number of cases in stage IIIC, no statistical differences). The assessment of the clinical stage to a previously unattainable degree of accuracy using histological techniques for lymph node analysis is known as ultrastaging. More frequent implementation of the sentinel node procedure (part of which is ultrastaging) could contribute to an increased detection of lymph node metastases. The use of SLN mapping ensured high feasibility, safety, and accuracy in the assessment of node metastasis. Gao et al. [30], through a retrospective analysis, demonstrated that the pelvic lymph nodes dissected by laparoscopy were significantly smaller than those dissected via laparotomy, but there was no significant difference in overall survival. The shortened time of surgery and the reduced frequency of lymphedema strongly support the concept of applying the sentinel node procedure in patients with high-risk endometrial cancer [31]. However, as confirmed in our study, sentinel node surgery significantly reduces hemoglobin loss in patients with endometrial cancer compared with other lymphadenectomy techniques.

## 5. Limitations

This study can be referred to as a developmental study because it was carried out on a small group of patients. This study is limited by the use of a historical control group treated by laparotomy. The obtained results are greatly influenced by the local experience and skills of individual operators as well as by the equipment at facilities. To confirm our results, a study should be carried out on a much larger group of patients, preferably multicenter in nature, including centers with different volumes and experience of operators. The study included only patients with endometrioid cancer. The analysis was focused on the trends and correlations of individual indicators and treatment results.

## 6. Conclusions

The frequency of complications occurring in patients treated with traditional or laparoscopic methods does not significantly differ. The course of surgical treatment using the classical method is associated with a significantly greater loss of hemoglobin and decisively longer hospitalization time. Abdominal obesity is a significant risk factor in the treatment of endometrial cancer. Obesity is associated with longer surgery duration and hospitalization, as well as greater blood loss in both laparoscopic and classic treatments. The crucial obesity-related parameters influencing the treatment of endometrial cancer are the parameters of abdominal obesity: waist circumference and WHR. The extent of lymphadenectomy is an important risk factor in hemoglobin loss. The sentinel node procedure is associated with the lowest hemoglobin loss during surgical treatment of endometrial cancer compared with other methods of lymphadenectomy. This study is limited by the use of a historical control group treated by laparotomy. However, the percentage of patients treated with laparotomy is still significant in Poland and at other centers around the world [32,33]. In this study, methods are reported that were used in the surgical treatment of endometrial cancer, which improved in-hospital outcomes, especially among obese patients.

## Figures and Tables

**Figure 1 jcm-10-00429-f001:**
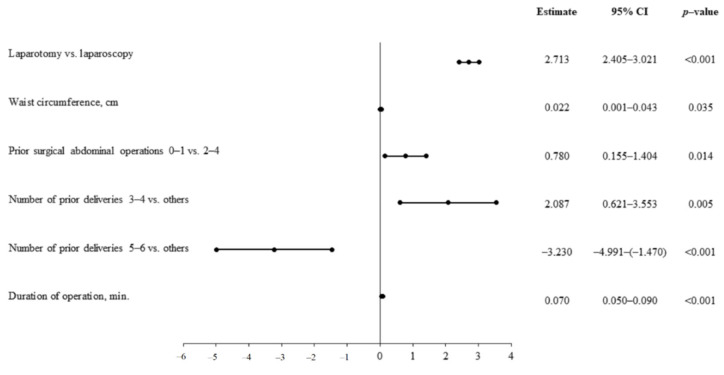
Predictors of the duration of hospitalization—multivariate linear regression analysis.

**Figure 2 jcm-10-00429-f002:**
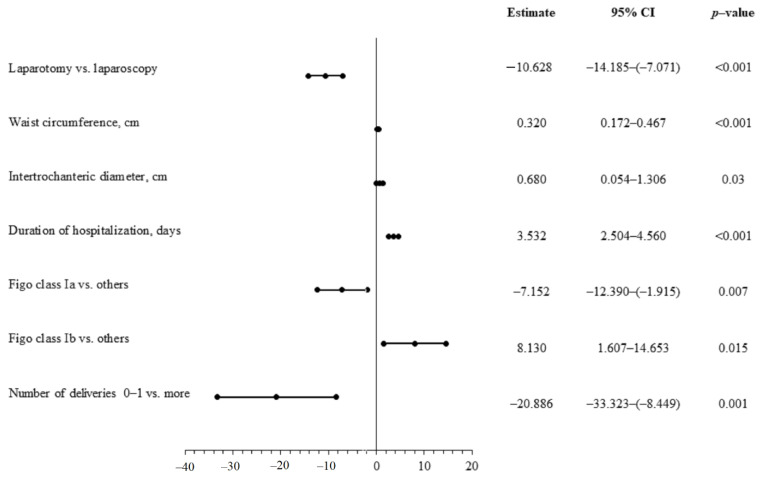
Predictors of the duration of operation—multivariate linear regression analysis.

**Figure 3 jcm-10-00429-f003:**
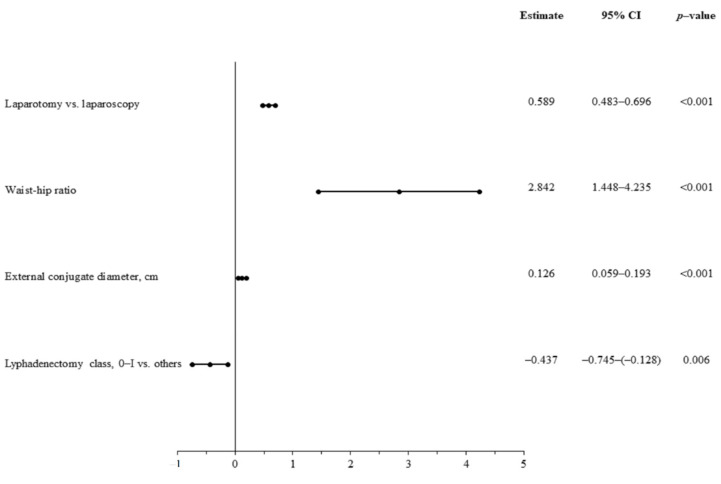
Predictors of the extent of hemoglobin loss—multivariate linear regression analysis.

**Table 1 jcm-10-00429-t001:** Characteristics of patients treated for endometrial cancer (laparoscopy vs. laparotomy)—anthropometric indices.

Variables	Total *n* = 145	Laparotomy *n* = 70	Laparoscopy *n* = 75	*p*-Value
Age, years	61.8 ± 9.5	61.3 ± 10.4	62.2 ± 8.7	0.55
Height, cm	160.9 ± 5.3 161 (158 ÷ 164)	160.1 ± 5.6 160 (157 ÷ 164)	161.5 ± 4.9 162 (158 ÷ 164)	0.22
Weight, kg	83.4 ± 19.8	82.6 ± 19.4	84.1 ± 20.3	0.65
Body mass index, kg/m^2^	32.2 ± 7.6 31 (26.4 ÷ 36.2)	32.2 ± 8.0 30.6 (26 ÷ 37.8)	32.1 ± 7.3 31 (27.2 ÷ 34.8)	0.73
Waist circumstance, cm	107.3 ± 18.6	105.3 ± 18.4	109.2 ± 18.8	0.24
Hip circumstance, cm	113.5 ± 18.0 111 (102 ÷ 125)	112.5 ± 15.5 109.5 (100.5 ÷ 125)	114.4 ± 20.2 112 (104 ÷ 125)	0.24
Waist-to-hip ratio	0.93 ± 0.1 0.94 (0.9 ÷ 0.97)	0.92 ± 0.1 0.92 (0.88 ÷ 0.97)	0.94 ± 0.1 0.95 (0.91 ÷ 0.97)	0.04
Obesity, Body mass index >30 kg/m^2^	84 (57.9)	37 (52.8)	47 (62.7)	0.24
External conjugate diameter, cm	22.0 ± 1.4 22 (21 ÷ 23)	22.2 ± 1.5 22 (21 ÷ 23.2)	21.9 ± 1.4 22 (21 ÷ 23)	0.21
Interspinosus diameter, cm	27.1 ± 2.0 27 (26 ÷ 28)	26.9 ± 2.1 27 (25 ÷ 28)	27.3 ± 1.9 27 (26 ÷ 28)	0.15
Intercristal diameter, cm	32.5 ± 2.8 33 (30 ÷ 34)	32.3 ± 3.0 33 (30 ÷ 34.2)	32.6 ± 2.6 33 (31 ÷ 34)	0.56
Intertrochanteric diameter, cm	35.2 ± 3.5 35 (35 ÷ 37)	35.3 ± 2.1 35 (34 ÷ 37)	35.1 ± 4.4 35 (35 ÷ 37)	0.15

**Table 2 jcm-10-00429-t002:** Characteristics of patients treated for endometrial cancer (laparoscopy vs. laparotomy)—clinical characteristics.

Variables	Total *n* = 145	Laparotomy *n* = 70	Laparoscopy *n* = 75	*p*-Value
Prior abdominal surgery	0.61 ± 0.7	0.65 ± 0.8	0.57 ± 0.6	0.54
Number of prior abdominal surgeries:				
0	71 (49.6)	34 (50)	37 (49.3)	0.92
1	61 (42.7)	28 (41.2)	33 (44)	0.62
2	8 (5.6)	3 (4.4)	5 (6.7)	0.53
3	2 (1.4)	2 (2.9)	0 (0)	0.14
4	1 (0.7)	1 (1.5)	0 (0)	0.29
Number of prior deliveries	2.4 ± 1.2	2.6 ± 1.2	2.2 ± 1.2	0.13
Most frequent concomitant diseases				
Diabetes mellitus	57 (39.3)	32 (45.7)	25 (33.3)	0.17
Arterial hypertension	67 (46.2)	33 (47.1)	34 (45.3)	0.82
Chronic obstructive pulmonary disease	16 (11)	8 (11.4)	8 (10.7)	0.88
Bronchial asthma	7 (4.8)	1 (2,4)	6(8)	0.05
Hypothyreosis	27 (18.6)	10 (4.3)	17 (22.7)	0.19
Chronic pancreatitis	11 (7.6)	3 (4.3)	8 (10.7)	0.13
Heart failure	9 (6.2)	5 (7.1)	4 (5.3)	0.65
Coronary artery disease	2 7 (18.6)	17 (24.3)	10 (13.3)	0.09
Number of concomitant diseases	1.8 ± 1.6	1.8 ± 1.5	1.9 ± 1.6	0.53

**Table 3 jcm-10-00429-t003:** Characteristics of patients treated for endometrial cancer (laparoscopy vs. laparotomy)—procedural indices.

Variables	Total *n* = 145	Laparotomy *n* = 70	Laparoscopy *n* = 75	*p*-Value
Duration of operation, min.	88.8 ± 19.4	86.1 ± 20.4	91.4 ± 18.2	0.09
Duration of hospitalization, days	8.8 ± 3.4 8 (6 ÷ 12)	11.3 ± 3.1 12 (8.7 ÷ 13)	6.5 ± 1.7 7 (5 ÷ 8)	<0.001
Class of lymphadenectomy	1.4 ± 0.8 1 (1 ÷ 2)	1.1 ± 0.9 1 (0 ÷ 2)	1.6 ± 0.7 2 (1 ÷ 2)	0.007
Class of lymphadenectomy:				
0	24 (16.5)	23 (32.8)	1 (1.3)	<0.001
I	50 (34.5)	15 (21.4)	35 (46.7)	0.001
II	64 (44.1)	31 (44.3)	33 (44)	0.97
III	7 (4.8)	1 (1.4)	6 (8)	0.06
Procedure-related complications	10 (6.9)	5 (7.1)	5 (6.7)	0.91
Periprocedural hemoglobin loss, g/dL	1.6 ± 0.9	2.25 ± 0.9	1.0 ± 0.2	<0.001
Staging according to FIGO classification: (Ia-1, IB-2, II-3, III-A-4, IIIB-5, IIIC1-6, IIIC2-7, IVA-8, IVB-9)	2.1 ± 1.4 2 (1 ÷ 3)	2.0 ± 1.5 1 (1 ÷ 2.25)	2.3 ± 1.3 2 (2 ÷ 3)	0.01
Staging according to FIGO classification:				
IA	54 (37.9)	36 (51.4)	18 (24)	0.011
IB	52 (35.9)	17 (24.3)	35 (46.7)	0.005
II	22 (15.2)	9 (12.9)	13 (17.3)	0.45
IIIA	8 (5.5)	5 (7.1)	3 (4)	0.4
IIIB	4 (2.8)	0 (0)	4 (5.3)	0.05
III C1	2 (1.4)	1 (1.4)	1 (1.3)	0.96
IIIC2	0	0	0	0
IVA	3 (2.1)	2 (2.9)	1 (1.3)	0.6
IVB	0	0	0	0
Histological grading; (G1-1, G2-2, G3-3)	1.52 ± 0.7 2 (1 ÷ 3)	1.56 ± 0.7 1 (1 ÷ 2.25)	1.49 ± 0.6 2 (2 ÷ 3)	0.53
Histological grading:				
1	81 (55.9)	37 (52.9)	44 (58.7)	0.48
2	51 (35.2)	26 (37.1)	25 (33.3)	0.63
3	13 (9)	7 (10)	6 (8)	0.67

**Table 4 jcm-10-00429-t004:** Mutual relationships between duration of operation, duration of hospitalization, and hemoglobin loss with selected indices (Spearman’s correlations).

	Duration of Operation		Duration of Hospitalization		Periprocedural Hemoglobin Loss	
	*r*	*p*-Value	*r*	*p*-Value	*r*	*p*-Value
Age, years	0.11	0.16	0.01	0.86	−0.03	0.68
Height, cm	−0.07	0.37	−0.13	0.11	−0.11	0.15
Weight, kg	0.55	<0.001	0.29	<0.001	0.29	<0.001
BMI, kg/m^2^	0.58	<0.001	0.31	<0.001	0.32	<0.001
Waist circumference, cm	0.65	<0.001	0.29	<0.001	0.26	0.001
Hip circumference, cm	0.57	<0.001	0.22	0.007	0.23	0.005
Waist-to-hip ratio	0.44	<0.001	0.13	0.1	0.1	0.19
Intertrochanteric diameter, cm	0.43	<0.001	0.08	0.31	0.05	0.5
Interspinosus diameter, cm	0.35	<0.001	0.08	0.28	0.02	0.77
Intercristal diameter, cm	0.31	<0.001	0.08	0.31	0.13	0.11
External conjugate diameter, cm	0.2	0.01	0.3	<0.001	0.23	0.004

## Data Availability

The data presented in this study are available on request from corresponding author.

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
