# Peer review of "Risk Factors for Surgical Treatment of Endometrial Cancer Using Traditional and Laparoscopic Methods"

_jcm, 2021, doi:10.3390/jcm10030429_

Round 1

Reviewer 1 Report

Dr. Januszek and colleagues are to be commended on their evaluation of risk factors influencing surgical treatment of endometrial cancer. The authors find a number of factors associated with worse perioperative outcome. I have several comments.

1) Introduction: The authors review the epidemiology of endometrial cancer, the association between endometrial cancer and obesity and the impact of minimally invasive surgery on the management of endometrial cancer. The authors need to focus the introduction on the objective (specific aim) of the study.

2) Introduction: Does the investigation appropriately address the stated specific aim "to determine factors having the most significant impact on the course of surgery in endometrial cancer and consequently influencing the choice of surgical method"? The investigation evaluated 75 patients treated by laparoscopy in 2019 and used a historical control of 70 patients treated by laparotomy in 2011. The authors state that the percentage of patients treated with laparoscopy increased substantially between 2011 and 2019. Were differences (especially preoperative) seen between the two groups a result of differences in the year of treatment rather than the surgical approach. The authors need to either change the specific aim of the investigation or address the limitations of a historical control.

3) Material and Methods. The authors collect data on a number of factors related to body habitus such as weight, height, body mass index, waist-hip ratio and pelvic bone dimensions. These factors are correlated. How does the inclusion of multiple correlated factors affect the results of multiple regression modelling? Table 4 presents correlation between different factors and duration of operation, duration of hospitalization and periprocedural hemoglobin loss. How are the different body habitus factors correlated? Weight to BMI, etc. Should some of the factors be removed from the modelling.

4) Material and Methods. Pelvic meters are utilized to determine pelvic dimensions and are often utilized for obstetrics. It is not unreasonable to include this data in the investigation. However, the authors should briefly provide the rationale for including the data in the study. Does the data correlate with the presence of endometrial cancer, surgical approach or body habitus? Why is this data included in the investigation?

5) Results. General characteristics. Table 1. Waist to hip ratio is significantly different between groups p=0.04. However, the difference is 0.92 versus 0.94. Although this is statistically significant is it clinically significant. Given the use of a historical control, does the evaluation compare laparotomy versus laparoscopy or compare 2011 versus 2019?

6) Results. Patient characteristics. Table 2. Same as above. Given the use of a historical control, does the evaluation compare laparotomy versus laparoscopy or compare 2011 versus 2019? Additionally, is a p value of 0.5 significant. The authors state, "The p-value of <0.05 was considered statistically significant." The formatting for the table needs to be improved. The rows for "number of prior abdominal surgeries" do not line up. Also the use of labels 0, -1, -2, -3, -4 could be improved. Would simply use 0, 1, 2, 3, 4.

7) Results. Tumor staging and grading. The extent of lymphadenectomy is also influenced by the use of a historical control. Were sentinel nodes performed at the authors institution in 2011? Do the authors have the ability to perform sentinel via laparotomy. The differences in lymphadenopathy appear to be related limitations inherent to the surgical approach. How does this affect surgical metrics? Duration of surgery is not simple comparing laparoscopy versus laparotomy but also sentinel node versus lymphadenectomy. Same with blood loss. Duration of hospital stay is 6.5 days for the laparoscopy group and 11.3 days for the laparotomy group. These durations appear longer than often seen in the literature. The authors should address. The rows for "staging according FIGO classification" do not line up. Also the use of labels 0, -1, -2, -3, -4, etc could be improved. Would simply use IA, IB, II, IIIA, IIIB, etc. For the comparison of all stages p = 0.01, what is the reference? IA? The first row for staging has 1 subject. What stage does this represent? Laparotomy has 50% stage IA, 24% stage IB and 12% stage II. Laparoscopy has 24% stage IA, 46% stage IB and 17% stage II. As above is this related to surgical approach or another factor? Surgical approach shouldn't affect depth of myometrial invasion, cervical involvement or involvement vagina and parametrium. I interpret the table to report 2 patients with stage IIIC disease. What is the average number of lymph nodes for patients with lymphadenopathy? What percentage of patients in the sentinel lymph node group mapped?

8) Discussion. Limitations. Would include use of a historical control as a limitation.

9) Conclusions. The study is limited by the use of a historical control. I would suggest it would be more interesting to report the methods utilized to increase laparoscopic management of endometrial cancer over the course of 8 years.  How the management of obese patients has been altered and whether the management of obese patients have improved.

Reviewer 2 Report

this is a discrete article adding data to prexisting literature on the topic, some comments are listed below:

Materials and methods:
Where the study was conducted? Specify the context and surgeon’s skills.
It seems that you obtain endometrial tissue by dilation and curettage (line 73). This method was the
principal method of investigation in the past. Several publications have reported that the accuracy of
curettage is limited, citing false-negative rates as high as 10%.
You never mention diagnostic methods used for staging cancer nor to what histological type of endometrial
cancer you refer. Considering that in tables you classified patient according to histological grading, I assume
that you included only endometriod histotypes however in line 92 you mention omentectomy for serous
and carcinosarcoma. Please clarify this aspect.
Results:
Line 150 do not use abbreviations without first defining them.
Table 3: what do you classify as a stage 0 FIGO?
Paragraph 3.6: you consider laparoscopy as an independent predictor of longer operative time, without
never considering or mentioning surgeon’s experience. Please precise.
Discussion:
Line 228 the two affirmations of FIGO staging are in contradiction with each other.
The description of study endpoints is quite extensive and you already described all these correlation in the
results. It would be better to summarize this part, focusing on why the outcomes and findings of this study
are clinically relevant.
Limitations:
What do you mean with developmental study?
Small grammar corrections: line 33 “abdominal obesity is AN important”, line 208 choose one between
significant and important. It is preferable any time to associate other author citations with the
corresponding bibliographic referral (example line 258, line 261).

Round 2

Reviewer 1 Report

Appreciate the authors' revision of the manuscript.  The changes enhance the quality of the article.  One last comment, I would recommend editing the abstract to align with the changes to the body of the manuscript. 

The abstract should mention that investigation evaluated 75 patients treated by laparoscopy in 2019 and used a historical control of 70 patients treated by laparotomy in 2011. Also, the last sentence suggests that sentinel nodes are associated with better outcomes than other approaches.  As sentinel nodes were only performed in patients undergoing laparoscopy, both laparoscopy and sentinel nodes probably contribute to better outcome.  I suggest concluding the abstract in a similar fashion as the manuscript.  Laparotomy continues to be utilized frequently in the management of endometrial cancer in Poland and elsewhere and adopting a minimally invasive approach is likely to be beneficial to patient outcome.  

Reviewer 2 Report

Dear authors

thank you for your replies

however to improve quality of the manuscript please add, comment and compare relevant studies on this topic 10.1016/j.jmig.2015.08.007

10.1016/j.jmig.2014.07.014

10.1016/j.ygyno.2015.09.020
